# Construction of Co,N-Coordinated Carbon Dots for Efficient Oxygen Reduction Reaction

**DOI:** 10.3390/molecules27155021

**Published:** 2022-08-07

**Authors:** Mengying Le, Bingjie Hu, Meiying Wu, Huazhang Guo, Liang Wang

**Affiliations:** Institute of Nanochemistry and Nanobiology, School of Environmental and Chemical Engineering, Shanghai University, 99 Shangda Road, BaoShan District, Shanghai 200444, China

**Keywords:** carbon dots, structure regulation, Co,N-doped, electrocatalysis, oxygen reduction reaction

## Abstract

For the sake of the oxygen reduction reaction (ORR) catalytic performance, carbon dots (CDs) doped with metal atoms have accelerated their local electron flow for the past few years. However, the influence of CDs doped with metal atoms on binding sites and formation mechanisms is still uncertain. Herein, Co,N-doped CDs were facilely prepared by the low-temperature polymerization–solvent extraction strategy from EDTA-Co. The influence of Co doping on the catalytic performance of Co-CDs was explored, mainly in the following aspects: first, the pyridinic N atom content of Co-CDs significantly increased from 4.2 to 11.27 at% compared with the CDs, which indicates that the Co element in the precursor is advantageous in forming more pyridinic-N-active sites for boosting the ORR performance. Second, Co-CDs are uniformly distributed on the surface of carbon black (CB) to form Co-CDs@CB by the facile hydrothermal route, which can expose more active sites than the aggregation status. Third, the highest graphite N content of Co-CDs@CB was found, by limiting the current density of the catalyst towards the ORR. Composite nanomaterials formed by Co and CB are also used as air electrodes to manufacture high-performance zinc–air batteries. The battery has good cycle stability and realizes stable charges and discharges under different current densities. The outstanding catalytic activity of Co-CDs@CB is attributed to the Co,N synergistic effect induced by Co doping, which pioneer a new metal doping mechanism for gaining high-performance electrocatalysts.

## 1. Introduction

The serious energy crisis has been increasingly prominent in recent years, which promotes us to exploit new types of energy storage and conversion equipment [1,2,3]. For instance, zinc–air batteries have appeared in the past few decades and have unpredictable prospects [4]. The oxygen reduction reaction (ORR) plays an important role in rechargeable zinc–air batteries as a fundamental reaction. The Pt-based precious metal has been deemed as the most commonly used catalyst for the ORR. However, its extremely high price and rareness (only 37 ppb in the Earth’s crust) hinders potential large-scale applications. In addition, the laggardly ORR occurring at the cathode severely obstructs the output power density and operating efficiency of these renewable energy systems [5]. Therefore, exploring a clean, low-cost, and sustainable electrocatalyst with a superior performance can address the above issues and further benefit the entire energy industry.

Carbon nanomaterials were considered to be the candidate material with the most potential as an ORR catalyst for fulfilling the above demands owing to its rich source, high environmental compatibility, and catalytic activity [6]. Experimental and theoretical results have demonstrated that nitrogen (N)-doped carbon catalysts exhibit more excellent activity than other heteroatom-doped carbon materials for the ORR [7,8]. Generally, there are three main routes to synthesize N-doped carbon materials: First, pure carbon materials, such as graphene, carbon nanotubes, and carbon nanofiber, are treated at high temperatures (>900 °C) in a NH_3_ atmosphere [5,9,10]. Second, N-containing organic small molecules as precursors are directly pyrolyzed, such as melamine, urea, and thiourea. Last, specific structural polymeric substances similar to metal–organic frameworks (MOFs) are carbonated in a NH_3_ atmosphere or are intrinsically N self-doping in their structure [11]. Even though employing these methods can obtain nitrogen-doped carbon materials, there are still some matters to be resolved: (1) it is difficult to achieve a high N-doping content using conventional pyrolysis under NH_3_ atmosphere and uncontrollable for the N types; (2) organic small molecule precursors can confine some N types. However, small molecules are firstly gasified before carbonization because of the low vaporization temperature and discharge of the carrier gas. Therefore, the yield of targeted nanocarbon is low [12]. Additionally, the N-doping content of nanocarbon is unstable and suffers serious loss, because those small molecules are likely to decompose at high temperatures during the carbonization process; (3) the polymeric substance precursors with an intrinsic nature are manufactured at a low yield by using an expensive reagent at the multiplying steps. After further carbonization, the total yield of the obtained carbon is super-low and thereby significantly restricts their practical application, although their catalytic performance may be very superior. Wholly, the above-mentioned pathways cannot attain a high N-doping content and high productivity, so it is urgent to exploit a brand-new stratagem to solve these niduses.

In recent years, zero-dimensional carbon dots (CDs) have attracted more and more attention due to their outstanding features of low cost, being nontoxic, large surface area, and rich surface functional groups [13], which have become the rising star in carbon-based nanomaterials. Notably, the size of CDs is less than 10 nm [14], which displays an extraordinary side effect [15,16,17,18]. The marginal ratio of the side positions of the CDs is far beyond the other carbon materials [19,20,21]. Furthermore, most of the synthesis reactions for the CDs are mainly solvothermal or hydrothermal approaches, which easily realize functionalized and doped CDs [22,23,24]. Recent research showed that CDs have great reactivity and huge potential in the ORR, which is expected to solve the above problems [25,26]. Niu et al. prepared N-doped CDs decorated in a graphene oxide hybrid catalyst through a simple one-step synthesis method as a high-efficiency low-cost metal-free electrocatalyst towards the ORR, which exhibited improved electrical conductivity [27]. Mohamedazeem M. Mohideen et al. discussed the application of CDs/carbon nanotube nanostructures in a fuel cell and highlighted the progress and future potential for improving the ORR performance [28]. There is no denying that CDs have some advantages in the ORR; however, a gap in the catalytic activity relative to commercial Pt/C needs to be further narrowed by introducing more active sites. Additionally, considering the poor electrical conductivity of pure CDs, the introduction of a suitable support can not only make up for the poor electronic conductivity of pure CDs and then elevate the electron transfer rate but change the morphology and crystal structure of the composite, thus optimizing the electronic structure and providing active centers for the composite [29,30]. It is essential to exploit the effective design of the CDs with more active sites and pick a suitable carrier for constructing high-efficiency CD-based ORR electrocatalysts [31].

Recently, we reported the cobalt (Co) atom dispersed on a N-doped carbon nanosphere catalyst (denoted as ECo@D) using EDTA-Co as a precursor via a simple precursor-engineering method [32]. The obtained ECo@D performed an excellent ORR. This discovery inspired us to further reverse whether we can use the EDTA series compounds to prepare the CDs and obtain more excellent catalytic activity than that of ECo@D. Here, Co-CDs were prepared by the low-temperature polymerization–solvent extraction strategy using EDTA-Co as a precursor. Furthermore, Co-CDs were loaded on carbon black (CB) to implement a composite material (Co-CDs@CB) for the ORR. The significant enhancement in the ORR performance of Co-CDs@CB was principally attributed to the synergy of the Co-CDs and CB, where the Co-CDs could provide abundant active edges/sites for the ORR, and CB could act as a conductive substrate to facilitate the electron transfer and also protect the Co-CDs from agglomeration to maximize the utilization efficiency of each active site [13]. The zinc–air battery is expected to achieve the key objectives of the fast rechargeable battery proposed by the U.S. Department of Energy: >300 Wh kg^−1^, 75 kwh^−1^ USD, 80% charging capacity within 15 min, and operating in a wide temperature range, because the multi-electron redox reaction and intrinsic safety characteristics can simultaneously achieve high theoretical energy and power density [33,34]. The excellent ORR performance also lays the foundation for making a self-assembled zinc–air battery with good performance, since Co-CDs@CB as a cathode catalyst enhances the ORR reaction kinetics. This work paves the way for the development of CDs in the ORR.

## 2. Materials and Methods

### 2.1. Synthesis of Co-CDs, CDs and Co-CDs@CB

Briefly, 1.6 g EDTA-Co was placed in a quartz boat with an analytical balance and calcined in a tubular furnace at 350 °C for 2 h with a heating rate of 5 °C min^−1^ under N_2_, so as to form carbonized Co-CDs [35,36]. To further purify it, the product was grounded and suspended in 100 mL DI water and treated ultrasonically at room temperature for 2 h. Then, an 800-nm aqueous phase filter membrane was used to filter the upper brown solution to remove the nonfluorescent sodium salt. The clear solution A was obtained after filtration until no exchange occurred. Finally, the concentrated suspension was dried in an oven at 60 °C to get a Co-CDs powder. The CDs were prepared by a similar synthesis method. It just replaced the precursor with EDTA and changed the temperature from 350 °C to 250 °C. The prepared Co-CDs and CB are mixed according to the mass ratio of 1:0.15; after hydrothermal compounding at 150 °C for 2 h, then by washing, filtering, and air drying, Co-CDs@CB was obtained.

### 2.2. Electrochemical Measurements

All the electrochemical measurements of the as-synthesized samples were performed at room temperature and atmospheric pressure using a rotating disk electrode (RDE) at an Autolab electrochemical workstation (AUT88032) in a typical three-electrode system. The catalyst-coated glassy carbon electrode (GC) disk (Autolab Research Instrumentation, Metrohm., Herisau, Switzerland, 5.0 mm in diameter) was fabricated by casting as-prepared samples of ink as the working electrode, while a graphite rod and Ag/AgCl (saturated 3 M KCl) served as a counter electrode and a reference electrode, respectively, and all the potentials are converted into a normal hydrogen electrode (RHE) by the Nernst equation. In a typical process to prepare the catalyst ink, catalyst powder (5 mg) was evenly dispersed in a 1-mL mixed solution containing 700 µL DI water, 250 µL isopropanol, and 50 µL Nafion (5 wt%, Dupont). DI water:isopropanol:5 wt% Nafion by a volume ratio of 14:5:1 was added. Then, the mixed solvent was sonicated for 1 h to form a homogeneous ink. Subsequently, 10 µL of the above well-dispersed catalyst ink was drop-casted onto the surface of the GC electrode using a pipette and allowed to dry at ambient conditions to afford a mass loading of about 255 µg cm^−2^. Additionally, 10 µL 5 wt% Nafion was drop-casted after the dried layer of catalyst to prevent the soluble CDs or Co-CDs from falling off. The loadings of the catalysts were kept constant to avoid any variations related to sample loading onto GC-RDE. The dried electrodes were set aside for the next electrochemical measurements, including the cyclic voltammetry (CV) test and linear sweep voltammetry (LSV) tests.

### 2.3. Apparatus and Characterization

The morphology of the catalysts was analyzed by high-resolution transmission electron microscopy (TEM; JEOL JEM-2100F) operating at 100 kV. The crystal structures of the samples were characterized with X-ray powder diffraction (XRD, Rigaku D/max-2500 using CuKα radiation, λ = 1.5418 Å). FTIR spectra were obtained on a Bio-Rad FTIE spectrometer FTS165 over the range of 4000–450 cm^−1^. X-ray photoelectron spectroscopy (XPS) was performed on an AMICUS electronic spectrometer by SHIMADZU using a 300 W Al-K radiation source. The Raman spectra of Co-CDs@CB were collected using a 633-nm laser.

### 2.4. Preparation of Zinc–Air Battery

The air electrode is prepared by uniformly coating the catalyst ink prepared in the ORR process on carbon paper and air-drying it. A zinc plate was used as an anode, with 6 M KOH and 0.2 M Zn(C_2_H_3_O_2_)_2_·2H_2_O as the electrolyte. The mass load of the electrocatalyst is 1 mg cm^−2^. All the electrodes are assembled into a self-made zinc–air battery.

In this work, we used a 0.1 M KOH solution as the electrolyte for the ORR measurement. Before starting the test, the 0.1 M KOH electrolyte was pre-purged with pure and continuous oxygen bubbling for 30 min to make sure it was O_2_-saturated, and the O_2_ ensured purging throughout the test. CV measurements with a sweep rate of 50 mV s^−1^ were one of the methods used for evaluating the ORR activities. The other LSV experiments were conducted to evaluate the pathway mechanisms at various rotating speeds (400–3600 rpm) with a sweep rate of 10 mV s^−1^. Then, the LSV was scanned at a potential scan rate of 10 mV s^−1^ at 1600 rpm, and a constant potential was applied at 1.5 V vs. RHE to the Pt ring disk electrode, resulting in the electrooxidation of H_2_O_2_.

For the data analysis, all the measured potentials in this work were converted into a reversible hydrogen electrode (RHE) by the following Nernst equation:E (RHE) = E (Ag/AgCl) + 0.0591 pH + 0.197(1)

For analyzing the ORR kinetics and calculate electron transfer number (n), we used Koutecky–Levich (K–L) plots, as shown in the following equation:(2)$$\frac{1}{j}=\frac{1}{j_{k}}+\frac{1}{j_{L}}=\frac{1}{j_{k}}+\frac{1}{{B\omega }^{\frac{1}{2}}}$$
(3)$$B=0.2\mathrm{nF}{\left(D_{O_{2}}\right)}^{2/3}\upsilon^{-1/6}C_{O_{2}}$$
where j is the measured current density, and j_k_ and j_L_ correspond to the kinetic and diffusion-limiting current densities, respectively. Additionally, B represents the Levich constant, and ω is the angular velocity. In Formula (3), n is the overall electron transfer number in the ORR process, F is a fixed value, namely the Faraday constant (96,485 C mol^−1^), D represents the oxygen diffusion coefficient (1.9 × 10^−5^ cm^2^ s^−1^ in 0.1 M KOH electrolyte), ν expresses the kinetic viscosity of the electrolyte (0.01 cm^2^ s^−1^), and $C_{O_{2}}$ means the bulk oxygen concentration in 0.1 M KOH (1.2 × 10^−3^ mol L^−1^).

For analyzing the rotating ring-disk electrode (RRDE) measurements, the percentage of the peroxide species was derived from the current at the Pt ring disk electrode, and the electron transfer number (n) was determined based on the following equations:(4)$$H_{2}O_{2}\left(\%\right)=200\;\times \frac{I_{R}/N}{I_{D}{+I}_{R}/N}$$
(5)$$n=4\;\times \;\frac{I_{D}}{I_{R}{/N+I}_{D}}$$
where I_R_ and I_D_ represent the currents at the ring and the disk, respectively, and N expresses the ring collection efficiency (N = 24.9% in this article, which was determined by the manufacturer) of the RRDE (Autolab Research Instrumentation, Metrohm, Switzerland).

## 3. Results and Discussion

### 3.1. Formation Mechanisms of Co-CDs

The fabrication of the Co-CDs@CB catalysts was obtained by a two-step hydrothermal synthetic strategy, as depicted schematically in Figure 1. It is reported that doping heteroatoms and metal into the CDs could not only increase the number of catalytically active centers but also influence its charge distribution [37]. Inspired by the results, we herein synthesized Co and N-doped carbon dots (Co,N-CDs) with EDTA-Co as the raw materials. The CQDs doped with Co and N both exhibited excellent catalytic performances, thanks to the better synergy effect between the carbon layers and the cobalt nanoparticles [38]. On top of that, the doping of Co was realized by the calcination of the precursor containing Co, namely Na_2_[Co(EDTA)], in a tube furnace at 350 °C for 2 h at a heating rate of 5 °C min^−1^ under a N_2_ atmosphere. Through the one-step calcination method, we can obtain carbon dots containing Co, named Co-CDs. When the calcination temperature is lower than 250 °C, the color of the carbonized product is fuchsia, the same as the precursor, which indicates that the synthesis condition makes it hard to carbonize the precursor to form Co-CDs. Increasing the temperature ranging from 250 °C to 350 °C, their color changes from fuchsia to black, demonstrating the construction of Co-CDs [39]. We can easily obtain the colloid Co-CD solution after an ultrasonic treatment. In contrast, when the temperature is higher than 350 °C, the carbonization degree is high, and the microscale-level powers are formed. Therefore, the most suitable calcination temperature is 350 °C.

Although the calcination temperature is as high as 350 °C, the obtained Co-CDs exhibit a lowly graphitized structure, as shown in the XRD pattern (Appendix A). The phenomenon may be caused by the fact that a large number of organic molecular skeletons and abundant functional groups are retained in the cross-linking/pyrolysis carbonization process of the precursor. In addition, the prepared Co-CDs are further supported on CB (Co-CDs@CB) in the form of a hydrothermal to gain a high conductivity structure, which was confirmed by the XRD results (Appendix A). Appendix A shows a typical wide-angle XRD pattern of the Co-CDs@CB [40]. An intense diffraction peak at approximately 20–30° was indexed to the graphitic carbon (002) plane, and the other sharp peak at 40–50° was assigned to the (100) plane [41], indicating that the crystallinity of the composite was perfect [42,43]. Compared to the XRD pattern of pure Co-CDs, Co-CDs@CB displayed an additional (100) plane due to the high graphitic structure of synthesized Co-CDs@CB, which should be beneficial for enhancing the conductivity of the synthesized catalysts in the ORR process.

### 3.2. Effect of Co Doping on the Size of CDs

The morphology of the as-prepared catalysts of the CDs, Co-CDs, and Co-CDs@CB were characterized by transmission electron microscopy (TEM). It can be found that the CDs and Co-CDs have good dispersion on the ultra-thin copper grid. The average size of CDs is 4.5 nm (Figure 1a), which is slightly larger than that of Co-CDs (3.5 nm) (Figure 1b). The Co element in the precursor plays a vital role in suppressing the growth of Co-CDs and generates an obvious edge effect [44]. In Figure 1c and Appendix A, it is found that Co-CDs are uniformly distributed on the surfaces of random CB, which can expose more active sites than the aggregation status. The high-resolution TEM image and the corresponding FFT pattern (Figure 1d) exhibit clear, ordered lattice fringes (0.21 nm, d-spacing value/interlayer distance) of Co-CDs on the composite that are attributed to the (100) lattice plane of graphite, indicating that Co-CDs have good crystallinity, which was consistent with the XRD results (Appendix A).

### 3.3. Structural Characterization of Co-CDs@CB

The chemical bonds, element composition, and state of the catalysts were detected by Fourier-transform infrared spectroscopy (FTIR) and X-ray photoelectron spectroscopy (XPS). In the full spectrum of X-ray photoelectron spectroscopy in Figure 2a, the CDs show three main signals of C, N, and O, whereas the Co-CDs and Co-CDs@CB exhibit four main signals with the additional Co signal. The Co signal in the Co-CDs and Co-CDs@CB are relatively weak due to the ultralow Co content. In comparison to the Co signal of the Co-CDs, the signal of the Co element in the Co-CDs@CB is even less obvious, owing to the mixture of the large amount of CB in the Co-CDs. Figure 2b–d are the C1s, N1s, and O1s spectra of the catalysis samples. The carbon atom content gradually increases the trend from CDs and Co-CDs to Co-CDs@CB, and the carbon atom content in Co-CDs@CB achieves 90.54 at%, which can strengthen the conductivity capacity in the ORR [45]. In Figure 2b, the C1s spectra can be deconvoluted into three peaks of 284.8, 286, and 287.5 eV, corresponding to C-C/C=C, C-N/C-O, and C=O, respectively. Compared with CDs and Co-CDs, the atom content of C=O in Co-CDs@CB decreases due to the interference of the stronger C-C/C=C peak. Additionally, three peaks of pyridine N, pyrrole N, and graphite N in the N1s spectra can be divided (Figure 2c), which are centered at 398.2, 399.8, and 401.8 eV. In comparison to the CDs, the pyridinic N atom content of the Co-CDs significantly increases from 4.2 to 11.27 at%, which indicates that the Co element in the precursor is advantageous in forming more pyridinic-N-active sites for boosting the ORR performance [46]. Among them, the graphite N content of Co-CDs@CB is the highest, which is conducive to the enhancement of the limiting current density of the catalyst towards the ORR and agrees with their XRD results [47,48]. In Figure 2d, the O spectra can be separated into C=O and C-O at 531 and 532.8 eV. The C=O atom content of Co-CDs@CB is the lowest in the three samples, which corresponds to the results of the C1s spectra [49]. In whole, the Co-CDs can be uniformly loaded onto the CB in the process of hydrothermal recombination. It is more likely to be chemically bonded to the carbon matrix, which is conducive to the electron transfer between the Co-CDs and CB. Co exists in the form of Co^2+^ in CDs, and the doping of Co can improve the performances of CDs in the spectral and electrical properties [50]. Additionally, the introduction of Co into CDs endows Co-CDs with more pyridinic N-active sites, which is a benefit in the superior ORR performance of Co-CDs@CB.

As shown in Appendix A, the CDs and Co-CDs exhibit similar chemical bonds of O-H, N-H, C=O, C=C, C-N, and C-O, whose absorption peaks are located at 3397, 2929, 1633, 1429, 1368, and 1229 cm^−1^, respectively. Compared to the Co-CDs, the CDs show an extra C-OH-stretching vibration peak at 1292 cm^−1^, which may be due to the incomplete carbonization of the precursor that leads to the residue. Meanwhile, we can find five characteristic absorption peaks for Co-CDs@CB in Appendix A. The narrow peaks at 3594, 3042, 1740, 1546, and 1259 cm^−1^ are related to the O-H, N-H, C=O, C=C, and C-O-stretching vibration peaks, respectively. It is necessary to point out that the C-N bond is dispersed during the compounding process, which can be attributed to the Co-doped one and can be inferred that the doped atoms are most likely to combine with the CDs at the edge.

### 3.4. ORR Electrocatalytic Performance of Catalysts

Cyclic voltammetry (CV) and Linear sweep voltammetry (LSV) were measured to access the electrocatalytic ORR activities of the as-prepared materials. Figure 3a shows the CV curves of the five materials. The positive reduction peak potential (Ep) on Co-CDs@CB was 0.8282 V, which dramatically surpassed those of the CDs (0.6547 V), Co-CDs (0.6938 V), and CB (0.5523 V). The LSV curves (Figure 3b) revealed that the Co-CDs@CB catalyst manifested excellent ORR activity at the CDs, Co-CDs, CB, and CDs@CB due to the largest onset potential (E_onset_), highest halfwave potential (E_1/2),_ and highest limiting current density (j_L_). Specifically, the E_onset_ (0.9895 V) and E_1/2_ (0.8430 V) of Co-CDs@CB were significantly superior to those for the CDs (0.7384 and 0.4653 V), Co-CDs (0.7848 and 0.5945 V), CB (0.7089 and 0.5454 V), and CDs@CB (0.95 and 0.85 V). Compared with the E_onset_ (0.9962 V) and E_1/2_ (0.8657 V) of commercial PtC, it also has a strong competitive advantage. These results demonstrate that the introduction of Co-doping into the Co-CDs@CB electrocatalyst can provide vital contributions in gaining a higher ORR activity [51]. Meanwhile, Co-CDs@CB also delivers an j_L_ of 4.09 mA/cm^2^, which is superior to CDs (2.45 mA/cm^2^), Co-CDs (2.02 mA/cm^2^), CB (2.12 mA/cm^2^), and CDs@CB (0.95 mA/cm^2^), reflecting the outstanding mass transfer performance of Co,N-coordinated Co-CDs@CB architectures. Moreover, the excellent ORR activity of Co-CDs@CB can be further evidenced by the lowest Tafel slope (56 mV dec^−1^) (Figure 3c), illustrating the fastest ORR kinetics.

Apart from superior catalytic activity, we measured and drew the Koutecky−Levich (K−L) plots, the assessment standard of catalytic selectivity. As shown in Figure 3d,e, the J_L_ increases with the promotion of the rotating speed. Meanwhile, the number of transferred electrons is 3.88–3.90, which conforms to the first-order-dynamic kinetics [52,53]. It is confirmed that this reaction process is a four-electron reaction path. In addition, the rotating ring disk electrode (RRDE) test is further achieved to obtain the electron transfer number (n) and yield of H_2_O_2_ (%) of Co-CDs@CB (Figure 3f). The result certified a 4e pathway during the ORR and a few amounts of H_2_O_2_ of about 2%, which was consistent with the RDE measurements, suggesting the fastest ORR kinetics and highest energy efficacy of Co-CDs@CB among all the samples.

### 3.5. Application in Rechargeable Zinc–Air Battery

Owing to the superior ORR reaction kinetics, the Co-CDs@CB was used as a catalyst to build a liquid zinc–air battery (ZAB), in which the foam nickel–carbon paper composite with Co-CDs@CB as the air cathode and a Zn sheet as the anode. As shown in Figure 4a, the Co-CDs@CB-loaded battery exhibits a high open-circuit voltage of 1.37 V. Additionally, Figure 4b shows that ZAB loaded with Co-CDs@CB can deliver a capacity of 593 mAh g^−1^ at 10 mA cm^−2^ (normalized to the mass of the zinc electrode), corresponding to an energy density of 711 Wh kg^−1^. The charge and discharge voltage cycles at the current density range from 10 to 100 mA cm^−2^, shown in Figure 4c. Even after many cycles, the battery with Co-CDs@CB has a high emission plateau at different current densities. Then, the self-assembled battery was electrically cycled to test its chargeable property, which retained its balance at a current density of up to 10 mA cm^−2^. It is worth noting that, after more than 180 h of cycling, the voltage increase of the battery is very small, only 50 mV. According to the charge–discharge cycles at the current density of 10 mA cm^−2^ in Figure 4d, we believe in the excellent long-term stability of Co-CDs@CB as a Zn–air battery catalyst. These results lead to the conclusion that Co-CDs@CB can offer great potential in the application of zinc–air batteries.

## 4. Conclusions

In summary, Co,N-doped CDs were facilely prepared by the low-temperature polymerization–solvent extraction strategy from EDTA-Co. Based on the morphology measurements from TEM, the Co element in the precursor plays a vital role in suppressing the growth of Co-CDs and generating abundant active edge sites. Furthermore, the pyridinic N atom content of the Co-CDs significantly increased from 4.2 to 11.27 at% compared with the CDs, which indicates that the Co element in the precursor is advantageous in forming more pyridinic-N-active sites for boosting the ORR performance. In addition, Co-CDs are uniformly distributed on the surface of the CB to form Co-CDs@CB by the facile hydrothermal route, which can expose more active sites than the aggregation status. It is more likely to be chemically bonded to the carbon matrix, which is conducive to the electron transfer between the Co-CDs and CB. Additionally, the highest graphite N content of Co-CDs@CB was found to enhance the limiting current density of the catalyst towards the ORR. The Co-CDs@CB serving as an outstanding ORR catalyst exhibited an E_p_ of 0.83 V, an E_onset_ of 0.99 V, an E_1/2_ half-wave potential of 0.84 V, and a limiting current density of 4.09 mA/cm^2^, respectively. Meanwhile, a complete four-electron ORR pathway was revealed by the K–L equation and RRDE measurements. The great enhancement in the ORR performance of Co-CDs@CB primarily lies in the synergy of the Co-CDs and CB, where the Co-CDs could provide abundant active edges/sites for electrocatalytic reactions, and CB could act as a conductive substrate to facilitate the electron transfer and also protect the Co-CDs from agglomeration to maximize the utilization efficiency of each active site. Through the measurement of zinc–air battery, we found that Co-CDs@CB has excellent performance as an air electrode catalyst. Even at a high current density of 100 mA cm^−2^, it has a stable discharge voltage. This work not only provides a simple strategy for designing CDs but also opens novel horizons for applying CDs for energy conversion and storage.

## Data Availability

The data can be made available upon reasonable request.

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
