# Peer review of "Construction of Co,N-Coordinated Carbon Dots for Efficient Oxygen Reduction Reaction"

_molecules, 2022, doi:10.3390/molecules27155021_

Round 1

Reviewer 1 Report

Thanks for the invitation to review the manuscript.  In the manuscript authors prepared Co based carbon dots to use it as ORR catalyst and further they demonstrated Zn-air battery. The manuscript can be published but require major revision.

1. Introduction should be more general. Authors should talk more about air battery system and why Zn-air battery has advantage over other systems. Recent article on metal air battery should be included Small, 2021, 17, 1903854; Adv Sci 2020, 7, 1902866 etc.

2. Is the electrolyte used for Zn air mixture of KOH and Zinc acetate?

3. Authors have talked about Pt/C but they did not compare their material with Pt/C.

4. The H2O2 yield of the ORR process should be detected to estimate the number of electron transfer. The outdated KL equation should not be used.

5. The ORR performance curve is not normal and there is no limiting current.

6. Since, authors demonstrated rechargeable Zn-air battery, they should provide OER data as well. OER is usually not good in carbon materials.

7. The charge discharge overpotential in Zn air battery is very high which is biggest disadvantage of this material to be used in Zn -air battery. Also, why authors did not account air electrode to measure the capacity of device.

Author Response

Open Review 1

Comments and Suggestions for Authors

Thanks for the invitation to review the manuscript.  In the manuscript authors prepared Co based carbon dots to use it as ORR catalyst and further they demonstrated Zn-air battery. The manuscript can be published but require major revision.

Response: Thanks for your positive comment.

  1. Introduction should be more general. Authors should talk more about air battery system and why Zn-air battery has advantage over other systems. Recent article on metal air battery should be included Small, 2021, 17, 1903854; Adv Sci 2020, 7, 1902866 etc.

Response: Thank you for your good advice. We cited the important document as refs. 33 and 34. Besides, we have added a deep description of the Zn-air battery in the revised manuscript: The zinc-air battery is expected to achieve the key objectives of the fast rechargeable battery proposed by the U.S. Department of Energy: > 300 wh kg-1, 75 k Wh-1 USD, 80% charging capacity within 15 minutes, and operating in a wide temperature range because the multi-electron redox reaction and intrinsic safety characteristics can simultaneously achieve high theoretical energy and power density.

  1. Is the electrolyte used for Zn air mixture of KOH and Zinc acetate?

Response: As you expected, we used the mixed solution of 6 M KOH and 0.2 M Zn(C2H3O2)2·2H2O as electrolyte.

  1. Authors have talked about Pt/C but they did not compare their material with Pt/C.

Response: Thank you for your good advice. We have added the LSV curve of Pt/C as discussed it in Figure. 3b in the revised manuscript:

  1. The H2O2 yield of the ORR process should be detected to estimate the number of electron transfer. The outdated KL equation should not be used.

Response: We employed the rotating ring disk electrode (RRDE) test for obtaining the electron transfer number (n) and yield of H2O2 (%) (Figure 3f), and finally confirmed that this reaction process is a four-electron reaction path.

  1. The ORR performance curve is not normal and there is no limiting current.

Response: Thank you for your good question. We conclude that the Co-CDs@CB also delivers a jL of 4.09 mA/cm2, which is superior to Co-CDs (2.02 mA/cm2) and CDs@CB (0.95 mA/cm2), reflecting the outstanding mass transfer performance of Co,N-coordinated Co-CDs@CB architectures.

  1. Since, authors demonstrated rechargeable Zn-air battery, they should provide OER data as well. OER is usually not good in carbon materials.

Response: Thank you for your good advice. Regrettably, due to the Omicron epidemic in Shanghai, our lab is still closed, so we cannot supply the experiment for comparison. We hope to get your understanding of our special situation.

  1. The charge discharge overpotential in Zn air battery is very high which is biggest disadvantage of this material to be used in Zn -air battery. Also, why authors did not account air electrode to measure the capacity of device.

Response: The air electrode structure is relatively simple, and the existing one usually has a single-layer structure. Due to such a simple structure, the internal resistance of the air electrode is large, the permeability and adsorption rate of oxygen cannot be guaranteed, and the performance of the electrode is greatly reduced. More importantly, the gas-liquid-solid three-phase interface in the air electrode is unstable, and the liquid gradually permeates and floods the solid phase area, which eventually leads to the liquid permeating the whole electrode. This series of problems leads to the long-term and stable operation of metal-air batteries.

Reviewer 2 Report

This manuscript describes a facile fabrication of CDs with high pyridine contents using Co-EDTA complex. This method is quite simple and effective. Moreover, the Co-CDs@CB shows outstanding performance for ORR that was described as air electrodes to manufacture high-performance zinc-air batteries. Hence, publication in Molecules is recommended.

Author Response

Open Review 2

Comments and Suggestions for Authors

This manuscript describes a facile fabrication of CDs with high pyridine contents using Co-EDTA complex. This method is quite simple and effective. Moreover, the Co-CDs@CB shows outstanding performance for ORR that was described as air electrodes to manufacture high-performance zinc-air batteries. Hence, publication in Molecules is recommended.

Response: Thanks for your positive comment.

Reviewer 3 Report

The authors synthesized doped carbon dots and examined their oxygen reduction reaction / zinc-air battery cathodes.  The results should benefit  electrocatalysts in batteries.  Several comments should be considered before publication on Molecules.

1) Several abbreviations are not defined before being used, such as "CB" on Line 17,  "RHE" on Line 127, "RRDE" on Line 172, "ZAB" on Line 311, etc.

2) There is no experimental proof that Co exists in the synthesized carbon dots.  Please provide Energy-dispersive X-ray spectrum or other data to support the existence of Co elements in carbon dots.

3) XRD patterns (shown in Figure 1S and Figure 2S) are not enough to conclude that the synthesized materials are carbon.  Please index FFT in Figure 1d or provide electron diffraction patterns of the dots to approve the carbon phase or provide other data.

4) What's the chemical state of Co in the carbon dots?  Co-N, Co-O, Co-C, or single atomic state?  If Co is doped, please provide experimental data.

5)  Line 268, "the doped atoms are most likely to combine with CDs at the edge": please give more experimental proof for the conclusion.  If the statement is true, Co should not dope into CD lattices but exist on edges of CDs.  

6) Line 270 should be Section 3.4; Line 306: "3.4." should be "3.5".

7) There are not enough experimental data to support some conclusions in Summary, such as "Based on the morphology 330 measurement from TEM, the Co element in precursor plays a vital role to suppress the 331 growth of Co-CDs and generate abundant active edge sites.", "Co-CDs are uniformly distributed on the surface of the CB".  Please provide more experimental data or revise these conclusions.

8) Some typos, such as a) Line 52, "Nitrogen-doped" should be "nitrogen-doped", b) Line 110, "dissolved in" should be "suspended in", c) Line 113, "concentrated solution" should be "concentrated suspension", d) Line 145, "AlK radi-" should be "Al-K radi-", e) Line 170, "and C means" should be "and C_{O2} means", f) Line 179, "C0-CDs" should be "Co-CDs", etc

9) Line 141, please double check the accelerating voltage of TEM is 70 kV, not 80 kV, nor 200 kV.

Author Response

Open Review 3

Comments and Suggestions for Authors

The authors synthesized doped carbon dots and examined their oxygen reduction reaction / zinc-air battery cathodes.  The results should benefit electrocatalysts in batteries.  Several comments should be considered before publication on Molecules.

Response: Thanks for your positive comment.

  • Several abbreviations are not defined before being used, such as "CB" on Line 17,  "RHE" on Line 127, "RRDE" on Line 172, "ZAB" on Line 311, etc.

Response: Thank you for your elaborative check. We have added the abbreviations when there appeared the first time, which were as follows: carbon black (CB), normal hydrogen electrode (RHE), rotating ring-disk electrode (RRDE), zinc-air battery (ZAB).

  • There is no experimental proof that Co exists in the synthesized carbon dots. Please provide Energy-dispersive X-ray spectrum or other data to support the existence of Co elements in carbon dots.

Response: There is the Co signal in the XPS survey spectra of Co-CDs in Figure 2a.

.

  • XRD patterns (shown in Figure 1S and Figure 2S) are not enough to conclude that the synthesized materials are carbon.  Please index FFT in Figure 1d or provide electron diffraction patterns of the dots to approve the carbon phase or provide other data.

Response: Thank you for your good suggestion. The high-resolution TEM image and the corresponding FFT pattern (Figure 1d) exhibit the clear ordered lattice fringes (0.21 nm, d-spacing value/interlayer distance) of Co-CDs on the composite that are attributed to the (100) lattice plane of graphite. Your advice has been taken into consideration, and index FFT data can help better analysis. Regrettably, due to the Omicron epidemic in Shanghai, our lab is still closed, so we cannot supply the experiment for comparison. We hope to get your understanding of our special situation.

  • What's the chemical state of Co in the carbon dots?  Co-N, Co-O, Co-C, or single atomic state?  If Co is doped, please provide experimental data.

Response: We confirmed that Co was doped in CDs and finally formed composite Co-CDs@CB by XPS test (Figure. 2a). In addition, we have consulted and quoted relevant literature as refs. 37, 38, and 50. It is reported that doping heteroatoms and metal into the CDs could not only increase the number of its catalytically active center but also influences its charge distribution. Inspired by the results, we herein synthesized Co, N-doped carbon dots (Co, N-CDs) with EDTA-Co as raw materials. The CQDs doped with Co and N both exhibits excellent catalytic performance thanks to the better synergy effect between the carbon layers and the cobalt nanoparticles, and we thought that Co exists in the form of Co2+ in CDs, and the doping of Co in CDs can improve the performance of CDs.

  • Line 268, "the doped atoms are most likely to combine with CDs at the edge": please give more experimental proof for the conclusion.  If the statement is true, Co should not dope into CD lattices but exist on edges of CDs.  

Response: Thank you for your good suggestion. We have deleted this misleading statement.

  • Line 270 should be Section 3.4; Line 306: "3.4." should be "3.5".

Response: Thank you for your elaborative check. We have modified as follows: 3.4 ORR electrocatalytic performance of catalysts, 3.5 Application in a rechargeable zinc-air battery. Besides, we have checked the full text to make sure that there are no such errors.

  • There are not enough experimental data to support some conclusions in Summary, such as "Based on the morphology 330 measurement from TEM, the Co element in precursor plays a vital role to suppress the 331 growth of Co-CDs and generate abundant active edge sites.", "Co-CDs are uniformly distributed on the surface of the CB".  Please provide more experimental data or revise these conclusions.

Response: The average size of CDs is 4.5 nm, which is slightly larger than the average size of Co-CDs (3.5 nm) (Figure. 1a,b). We can infer that the doping of Co in the precursor inhibits the growth of Co-CDs. In addition, in Fig. 1c and Fig. S3, we found that Co-CDs are evenly distributed on the surface of random CB, which can expose more active sites than the aggregated state.

  • Some typos, such as a) Line 52, "Nitrogen-doped" should be "nitrogen-doped", b) Line 110, "dissolved in" should be "suspended in", c) Line 113, "concentrated solution" should be "concentrated suspension", d) Line 145, "AlK radi-" should be "Al-K radi-", e) Line 170, "and C means" should be "and C_{O2} means", f) Line 179, "C0-CDs" should be "Co-CDs", etc

Response: We have revised everything according to your good suggestions. Besides, we have checked the full text to make sure that there are no such errors.

  • Line 141, please double check the accelerating voltage of TEM is 70 kV, not 80 kV, nor 200 kV.

Response: Thank you for your elaborative check. We have checked the accelerating voltage of TEM is 100 kV.

Round 2

Reviewer 1 Report

Authors have revised the manuscript well as per reviewer suggestions. Therefore, it can be accepted for publication in its current form.